# Density Functional Theory and Density Functional Tight Binding Studies of Thiamine Hydrochloride Hydrates

**DOI:** 10.3390/molecules28227497

**Published:** 2023-11-09

**Authors:** Ewa Napiórkowska, Łukasz Szeleszczuk, Katarzyna Milcarz, Dariusz Maciej Pisklak

**Affiliations:** 1Department of Organic and Physical Chemistry, Faculty of Pharmacy, Medical University of Warsaw, Banacha 1 Str., 02-093 Warsaw, Poland; ewa.napiorkowska@wum.edu.pl (E.N.); katarzyna.milcarz5@gmail.com (K.M.); dariusz.pisklak@wum.edu.pl (D.M.P.); 2Doctoral School, Medical University of Warsaw, Żwirki i Wigury 81 Str., 02-093 Warsaw, Poland

**Keywords:** DFT, DFTB, GIPAW, CASTEP, hydrate, thiamine hydrochloride, ssNMR

## Abstract

Thiamine hydrochloride (THCL), also known as vitamin B_1_, is an active pharmaceutical ingredient (API), present on the list of essential medicines developed by the WHO, which proves its importance for public health. THCL is highly hygroscopic and can occur in the form of hydrates with varying degrees of hydration, depending on the air humidity. Although experimental characterization of the THCL hydrates has been described in the literature, the questions raised in previously published works suggest that additional research and in-depth analysis of THCL dehydration behavior are still needed. Therefore, the main aim of this study was to characterize, by means of quantum chemical calculations, the behavior of thiamine hydrates and explain the previously obtained results, including changes in the NMR spectra, at the molecular level. To achieve this goal, a series of DFT (CASTEP) and DFTB (DFTB+) calculations under periodic boundary conditions have been performed, including molecular dynamics simulations and GIPAW NMR calculations. The obtained results explain the differences in the relative stability of the studied forms and changes in the spectra observed for the samples of various degrees of hydration. This work highlights the application of periodic DFT calculations in the analysis of various solid forms of APIs.

## 1. Introduction

Most of the chemical compounds exist in a variety of crystal forms depending on the temperature, pressure, humidity, and solvents used during the crystallization process, which is the definition of the phenomenon known as polymorphism [1]. Polymorphic forms of active pharmaceutical ingredients (APIs) may differ in certain crucial characteristics, including solubility in water, dissolution rate, melting point, stability, tabletability, and others, which may ultimately affect the drug’s stability and bioavailability [2,3,4].

Hydrates, a subtype of solid solvates, are a distinct set of structures, although they share certain similarities with polymorphs. Once water molecules are incorporated into a compound’s crystal lattice, they modify the intricate H-bonding network [5]. Therefore, hydrates display a distinct structure when compared to corresponding anhydrates [6]. As a result, hydrates could differ from their anhydrous counterparts in terms of their physical and chemical characteristics [7]. Hydrates were once referred to as pseudo-polymorphs because of their similarities to polymorphs [8]. This term, however, is no longer considered to be the proper one when referring to hydrates [9].

Among the solid API solvates, hydrates are particularly interesting for several reasons. First of all, the water molecule stands out because of its special properties, which include its tiny size and ability to form interactions as both a donor and acceptor of H-bonding, occasionally concurrently; this makes it a crucial “building material” in the field of crystal engineering [10]. In addition, unlike the majority of other organic solvents, water is a safe and non-toxic substance. Finally, due to the air’s moisture content, spontaneous hydration may happen at any point during the manufacture or storage of an API, resulting in the development or phase transition of hydrates [11].

Non-stoichiometric hydrates, sometimes referred to as variable hydrates, are a unique type of hydrates in which the water content is constantly changeable as a function of the atmospheric water vapor pressure [12]. The water molecules in the non-stoichiometric hydrate lattice may occupy any or all of the predetermined places [13]. Such hydrate lattices frequently have channels or connected routes that allow water to easily enter or exit the lattice, causing the water content to continuously fluctuate as a function of the water vapor pressure.

The stability of both the API and the drug product can be affected by changes in hydration status. A formed drug may experience a chemical breakdown of one or more formulation components as a result of released water. Three outcomes are possible when water is removed from a drug’s crystal lattice: (1) the lattice structure is unaffected by the water removal, (2) the lattice collapses, or (3) the lattice packing is changed upon dehydration. Due to vacancies within the lattice, the product phase is frequently more reactive and less stable when the lattice structure is preserved following dehydration (outcome 1 above). Similar to outcome 1, the noncrystalline product phase is metastable when dehydration causes a collapsed lattice. Properties like the rate of dissolution can be impacted by changes in the physical form (outcome 3). Therefore, the process of dehydration and the subsequent release of water into the formulation might have a significant impact on overall drug stability [14,15].

As previously stated, the stability, solubility, bioavailability, and formulatability of solid-state APIs are influenced by their physical and chemical characteristics, which are the result of how molecules are arranged in the solid state. Therefore, it is intriguing from a strictly scientific perspective, as well as having significant practical significance in the pharmaceutical industry, to be able to precisely foresee and explain such qualities utilizing molecular modeling tools [16].

When it concerns solid APIs, molecular modeling techniques are typically used to forecast their physicochemical and structural characteristics, explain experimental findings, or predict the conditions necessary to produce novel forms of solid pharmaceutics in order to reduce the number of experiments or improve the experimental setup. Additionally, calculated properties like Nuclear Magnetic Resonance (NMR) shielding constants can make solid-state analysis much easier [17].

However, molecular modeling techniques that simulate a single molecule in vacuum or in solution were proven to be inadequate and erroneous as the distinctive properties of each solid form result from short- and long-distance intermolecular interactions [18]. While those “single molecules” approaches are often utilized effectively in other areas of pharmaceutical sciences, such as to examine drug–biomolecule interactions or predict the formation of complexes, alternative types of calculations should be used to study solid-state pharmaceutics. Those kinds of calculations, DFT and DFTB, under periodic boundary conditions, were performed in the present study.

Thiamine (Figure 1), also known as vitamin B_1_ and thiamin, is one of the most important vitamins and must be supplied with food [19]. In a properly balanced diet in healthy people, the effects of its deficiency are rarely observed because of its rich content in wholegrain products and an artificial enrichment of some food products with B-group vitamins. On the other hand, vitamin B_1_ is sensitive to UV radiation and high temperatures, especially in alkaline pH [20]. In addition, some drugs, e.g., chemotherapeutics, may inactivate thiamine [21], while others may reduce its absorption by competing for the transporter, such as metformin, used as the first choice drug in the treatment of diabetes [22]. All these factors may contribute to the occurrence of deficiencies of this vitamin, even in developed countries. Thiamine in a dose of 50 mg of thiamine hydrochloride (**THCL**) in tablets is on the List of Essential Medicines developed by the World Health Organization (WHO), which proves its importance for public health [23].

The object of this study, **THCL**, is hygroscopic and can occur in the form of hydrates with varying degrees of hydration, depending on the humidity. The anhydrous hydrochloride absorbs water and transforms into a non-stoichiometric hydrate (**NSH**) with a heterogeneous water content in the crystal, up to a compound with an equimolar ratio. In addition, **NSH** at room temperature and air humidity >53% can transform into a thermodynamically more stable form, i.e., hemihydrate (**HH**) [24]. Such transformation can start already at the stage of tablet production by wet granulation, which is used in the production of drugs from poorly tableting substances, such as **NSH**. The use of water in the granulation process causes the **NSH** to dissolve and then recrystallize into the more stable **HH**. As a result of an incomplete transformation, thiamine hydrochloride can be present in both forms: **NSH** and **HH**. During storage, the remaining part of **NSH** may be transformed into **HH**, while the addition of some excipients may slow down the transformation process [25]. The conversion of **NSH** into **HH** changes the properties of the tablets, e.g., an increase in hardness and an increase in the disintegration time of the tablet, which results in lower bioavailability of the drug [26].

Although experimental characterization of thiamine hydrates has been described in the literature [14,24,25,27,28,29], the questions raised in previously published works suggest that additional research and in-depth analysis of **NSH** and **HH** dehydration behavior are still needed. For example, Chakravarty et al. [28] speculated whether the respective physical stabilities of **NSH** and **HH** can be explained by the water-binding environment in the **NSH** and **HH** lattices. In their other work, they explicitly stated that “*Molecular modeling studies will be necessary to further understand the observed dehydration-induced SSNMR spectral changes.*” [27].

Inspired by those previously raised questions, we have decided to address the topic of thiamine dehydration using a variety of molecular modeling methods. Therefore, the main aim of this study was to characterize, by means of quantum chemical calculations, the behavior of thiamine hydrates and explain the previously obtained NMR spectroscopic results at the molecular level.

## 2. Results and Discussion

### 2.1. Crystal Structure Analysis

As stated in the introduction, **THCL** exists in a variety of forms, differing in the degree of hydration.

One of them, **NSH**, is characterized by variable water content in the crystal lattice up to an equimolar ratio. In extreme cases, this can lead to the formation of a monohydrate. The crystal structure of **NSH** in the form of monohydrate was deposited in CCDC under refcodes **THIAMC12** and **THIAMC14**. The authors of **THIAMC12** also deposited the structure of **NSH** after complete dehydration (refcode **UNEXOA**), which resulted in the formation of an anhydrous version of **NSH**.

Another solvate form of **THCL** is a stoichiometric hemihydrate (**HH**), whose structure has been deposited under refcode **WUWJAA**. This hemihydrate is a thermodynamically stable form. Its forced dehydration results in the collapse of the crystal lattice and sample amorphization; this is the reason why the crystal structure of the dehydrated form of **HH** has never been recorded before.

Chosen crystallographic information on the structures that we will be referring to in this study is presented in Table 1.

### 2.2. Crystal Structure Preparation

#### 2.2.1. NSH

As presented in Figure 2, there are four crystallographically equivalent water molecules in the unit cell of the monohydrate form of **NSH (THIAMC12)**, labeled as **UL** (upper left), **BL** (bottom left), **UR** (upper right), and **BR** (bottom right). As our aim was to simulate the gradual dehydration of this form, we decided to remove the water molecules one by one. Due to the symmetry of the system, the choice of one of the four water molecules to remove had no influence on the results of the calculations. Therefore, the **UL** molecule was removed, resulting in the formation of the 0.75 hydrate, named **MH3W**, due to the three water molecules left in the unit cell. However, the number of possible different versions of unit cells increased significantly when creating the hemihydrate form of **NSH**. While the number of permutations describing the possible options of removing two out of four water molecules present in the unit cell equals six, among those six structures, three crystallographically equivalent pairs existed. Finally, the structure of the 0.25 hydrate was obtained by leaving one water molecule in the unit cell. Again, due to the symmetry of the system, the choice of the molecule to be left had no influence on the results of the calculations. To clarify the process of structures’ generation, Table 2 was created.

#### 2.2.2. HH

As presented in Figure 3, there are four crystallographically equivalent water molecules in the unit cell of hemihydrate (**HH**) form of **THCL (WUWJAA)**, labeled as **UL** (upper left), **BL** (bottom left), **UR** (upper right), and **BR** (bottom right). As our aim was to simulate the gradual dehydration of this form as well, we have decided to remove the water molecules by one, similarly to **NSH**. Again, due to the symmetry of the system, the choice of one of the present four water molecules to remove had no influence on the results. Therefore, the **UL** molecule was removed, resulting in the formation of the 0.375 hydrate, named **HH3W**, due to the three water molecules left in the unit cell. However, the number of possible versions of unit cells increased significantly when creating the 0.25 hydrate from **HH**. Similarly to the **NSH**, the number of permutations describing the possible options of removing two out of four water molecules present in the unit cell equals six; among those six structures, three crystallographically equivalent pairs existed. Finally, the structure of the 0.125 hydrate was obtained by leaving one water molecule in the unit cell. Again, due to the symmetry of the system, the choice of the molecule to be left had no influence on the results of the calculations. To clarify the process of structures’ generation, Table 3 was created.

### 2.3. Crystal Structure Optimization

The results of the geometry optimization of structures described in Table 2 and Table 3 are presented in Table 4 and Table 5. For more facile comparison, chosen experimental structures (EXP) were also included in Table 4 and Table 5. For better clarity, the results of optimization are also presented in Figure 4 and Figure 5.

A good agreement has been observed between the corresponding experimental and computational results for both **NSH** and **HH**. What is interesting is that the difference between the theoretical value of the volume of **MH4W** and **EXP I** was found to be significantly lower than between **EXP I** and **EXP II** (3 Å^3^ versus 33 Å^3^); this is probably due to the temperature at which the SCXRD measurements were performed, namely 296 K and 173 K for **EXP I** and **EXP II**, respectively (Table 1). The structure recorded at a lower temperature is closer to the DFT-optimized one, as during the geometry optimization, the temperature was not included in the calculations. Also, as described in the introduction, the unit cell dimensions of **NSH** changed only slightly upon dehydration, with the exception of b length.

Even smaller changes in the unit cell dimensions have been observed for **HH**. In this case, the predictability of calculations was harder to assess due to the presence of solely one experimental structure of this form. According to the experimental results, any attempt to dehydrate the **HH** results in the collapse of the crystal lattice and sample amorphization. However, the results of geometry optimization from this work suggest that such structures could potentially exist; this can be explained by the fact that during geometry optimization, the influence of temperature is neglected, and the dynamic stability cannot be confirmed [30,31,32]. Therefore, we have conducted the ab initio molecular dynamics simulations (aiMD), which are described in detail in Section 2.6. However, before the analysis of aiMD results, some energetic aspects should be discussed, which is carried out in the next Section 2.4.

### 2.4. Energetic Considerations

A comparison of the structures with the same THCL: water ratio is presented in Table 6.

The results of the calculations are in very good agreement with the experimental observations. They show that the **HH4W** is more stable than any of the modeled hemihydrates based on the structure of **NSH**, **MH2W BLUR**, **MH2W ULBL**, and **MH2W ULUR** by a few kcal/mol, which was reported previously; this also explains the conversion of **MH2W** to **HH4W** upon dehydration of **MH4W**. The positive (+1.688 kcal/mol) value of the **Δ(HH-MH)** for the anhydrous “**0W**” structures is in agreement with the experimental findings, revealing that it is possible to obtain the crystalline **MH0W** while the complete dehydration of **HH0W** results in the complete crystal structure destruction and sample amorphization.

To compute the energy change upon dehydration, calculation of the energy of the single water molecule was necessary as the H_2_O is a product of such a reaction. While it is technically impossible to perform the calculations for nonperiodic systems in CASTEP, there is an alternative way to obtain such results, often called “molecule in the box” calculations. It consists of the creation of the unit cell, usually cubic, with a single molecule of interest inside it and unit cell lengths long enough to suppress any intermolecular interactions. Such a system is then optimized but with constrained unit cell dimensions. In this study, the cubic unit cell with equal lengths of 20 Å and a single water molecule inside it has been used to calculate the energy of H_2_O; this allowed us to determine the dehydration energies at each step, which are presented in Table 7.

Analysis of the values presented in Table 7 shows that, as expected, each dehydration step is endothermic for both **HH** and **MH**. Also, dehydration of **HH** requires more energy at each stage of this process; this is in agreement with the experimental results showing that dehydration of **MH** is significantly less demanding than **HH**. The lowest values observed in the last step for both **HH** and **MH** are probably due to the creation of a higher symmetry structure upon complete dehydration and reduction of Z’ to 1.

### 2.5. NMR Calculations

As stated in the introduction, **NSH** and **HH** have been extensively studied previously using solid-state NMR (ssNMR). However, for both those solvates the NMR experimentalists have found some observations that were not fully explained. Therefore, to understand, at the molecular level, the changes in the ^13^C ssNMR spectra of **NSH** and **HH** occurring upon dehydration, the GIPAW calculations of NMR chemical shielding constants have been conducted, and the results are presented below. Due to the differences in the atom numbering between the previously published works, in this work, the atom numbering presented in Figure 6 has been used.

#### 2.5.1. NMR Calculations of NSH

The results of the NMR calculations for the **NSH** of various degrees of hydration are presented in Table 8. As can be seen, the presence of water molecules has a major impact on the chemical shift (δ) value for most of the carbon atoms. For example, in C3, when the structure presents the monohydrate (**MH4W**), the values for all four atoms in the unit cell are close to 140.76 ppm (blue color), and the dehydration results in the increase of this value to 143.28 ppm (red color). Similarly, in **MH3W**, three “blue” (hydrated) values and one “red” (dehydrated) value were obtained, while for **MH1W**, one “blue” and three “red” values are present. Similar behavior has also been observed for C8, C1, C2, and C4.

Interestingly, major differences have been observed among the three structures presenting hemihydrate **MH2W**, which is well demonstrated by the SD values. For example, in C8, in the cases of **MH2W BLUR** and **ULUR**, the two distinct sets of values are observed, close to either 162.5 ppm (red) or 165 ppm (blue), which results in the high values of SD, over 1.2 ppm. On the contrary, in **MH2W ULBL**, chemical shifts for all four carbon atoms are averaged, and the SD equals only 0.27 ppm. Similar observations have been made for C9, C2 and C3.

Analysis of the changes in the experimental ^13^C CP MAS NMR spectra resulting from dehydration of NSH (Figure 7) revealed that this process has the greatest impact on the chemical shift values of C3, C6, C5, and C4.

For C3, the increase of δ has been observed both experimentally (1.9 ppm) as well as in the calculation’s results (2.5 ppm). Since in the spectrum of the 0.41 hydrate, the broadened peak of the averaged value of δ is observed instead of the two separate signals; this could indicate that **MH2W ULBL** is the dominant structure of the hemihydrate form, as in this one, the lowest value of SD for the calculated chemical shifts is observed (0.53 ppm vs. 1.96 ppm and 1.75 ppm for **ULUR** and **BLUR**, respectively). It should also be noted that among those three structures of **MH2W**, **ULBL** was the one with the lowest energy and, therefore, theoretically, the most stable one (Table 6). Similar observations can also be made for C4, C5, and C6 with even better accuracy between the experimental and theoretical increase of the δ upon dehydration. For example, for C4, the experimental one was 1.00 ppm, and the calculated one was 0.75 ppm.

For all of the carbon atoms, with the exception of C9, the sign of the calculated differences between the δ of the monohydrate and anhydrous forms (**MH0W-MH4W**) was in agreement with the experimental observations (**EXPII MH0W-EXPII MH4W**).

The NMR calculations also helped to properly assign the chemical shifts of C8 (C 2′) and C9 (C 4′). In the experimental spectrum (Figure 8), those two atoms were assigned to the broad peak at 163.20 ppm. According to the calculations results, the δ of C8 (163.95 ppm) is slightly higher than that of C9 (160.45 ppm). The experimentally observed averaging of this value, resulting in the overlapping of those two peaks, is probably a result of molecular dynamics.

#### 2.5.2. NMR Calculations of HH

The results of the NMR calculations for the **HH** of various degrees of hydration are presented in Table 9. It should be noted that in the only work in which the ^13^C ssNMR spectra of **HH** are being presented [27], the authors have not assigned any of the peaks to the particular carbon atom. In addition, the chemical shift values have not been provided either; therefore, the data presented in Table 9 in rows entitled “**EXP HH4W**” have been read by us directly from the spectra; this is why they have been rounded to either whole or half values, ±0.5 ppm.

As can be seen, similarly to the **NSH**, the presence of water molecules has a major impact on the chemical shift (δ) value for some of the carbon atoms. The authors of [27] have created a “*partially dehydrated*” **HH_2_** form of **HH**, “***HH***
*was dried at 60 °C in a bench-top freeze dryer (Unitop 400L, Virtis, Gardiner, NY) under reduced pressure (20–60 mTorr) for 7 days. This product phase will be referred to as **HH_2_**.*” They then recorded the ^13^C ssNMR spectra of both **HH** and **HH_2_** and found some differences, which are presented in Figure 9.

As noticed by the authors of [27], “*(…) partial dehydration of **HH** under low pressure for 7 days (**HH_2_**) resulted in a new peak at 157 ppm and the appearance of shoulders on the peaks at 148, 138, 65 ppm (…).*” Those researchers were speculating on the origin of those extra peaks and concluded that “*Molecular modeling studies will be necessary to further understand the observed dehydration-induced SSNMR spectral changes.*”

The results of the calculations, presented in Table 9, are in excellent agreement with the experimental observations. The largest difference of the calculated δ for the same atom, observed for **HH4W** and **HH0W**, was found for C1, and only for this atom did the difference in δ calculated for the “hydrated” and “dehydrated” structure exceed 1 ppm. Further, this was the only peak with direct splitting and evident separation observed in the experimental spectra (Figure 9). Also, the calculated δ was larger for C1 in the dehydrated structure (**HH0W**) than in the hydrated one (**HH4W**), which is also in agreement with the experimental observations.

To explain this observation made for the peak of C1, the authors of [27] have postulated two hypotheses. The first one stated that the loss of water causes changes in ^13^C–^14^N quadrupolar coupling. This hypothesis has been ruled out by the same authors based on the comparison of the spectra recorded at 300 MHz and 400 MHz spectrometers, as the differences in the width of signals were not significant. The second hypothesis was that the removal of water changes the physical location of the chloride ions within the crystal lattice. This hypothesis could not have been tested so far because of the lack of the experimental structure of the dehydrated form of **HH**.

It should be noted that in the structure of **HH**, two crystallographically inequivalent chloride ions can be found (Figure 10). The first one, Cl1, forms three intermolecular interactions with thiamine, two of them with H atoms of amine groups and one with the hydroxyl group. The second one, Cl2, forms one interaction with the H atom of water and one with the H atom of the thiazole ring of thiamine.

During the geometry optimization of the subsequently dehydrated structures of **HH** (Figure 11), we noticed that the RMSD of Cl1 was much higher than this of Cl2, which was caused by the fact that removing the water molecule that forms the intermolecular interaction with Cl2 causes this ion to move, and also results in the increase of the C1–H bond length; this consequently resulted in the increase of chemical shift of C1 and the presence of two peaks in the ^13^C ssNMR spectrum of **HH_2_** (Figure 9).

In the case of the peak at 148 ppm, the authors of [27] have noticed a slight downfield shift, and this was also observed in the calculations results for C11 as the increase of the δ by 0.4 ppm. The change of this magnitude did not result in the peak separation, only with the increase of its width.

For the peak at 138 ppm, the experimentalists have observed the formation of another peak at the right shoulder of the one present in **HH**, resulting from partial dehydration; this was also present in the calculation results for **C3**—the upfield change by 0.5 ppm was observed as the result of the dehydration.

No major changes in the values of δ computed for the other atoms, occurring as a result of dehydration, have been observed, which is also in agreement with experimental results.

It should also be noted that the differences in the values of δ found between the different versions of **HH2W**, namely **ULBL, BLUR, and ULUR**, within the same carbon atom, are much smaller than those observed for the **NSH**. Also, the difference between the highest and lowest energy forms of **HH2W**, **ULBL**, and **ULUR** is much lower than in the case of **MH2W** (Table 6)—0.188 versus 0.644 kcal/mol, respectively; this indicates that, unlike for **MH2W**, the various forms of **HH2W** are energetically similar and that the order in which the water molecules are removed from the crystal lattice is more random for **HH** than it was for **NSH**.

### 2.6. Molecular Dynamics Simulations

Despite the high accuracy in predicting the structure, energy, and NMR parameters, the ab initio CASTEP calculations were computationally too expensive to perform the Molecular Dynamics (MD) simulations using them. Therefore, for this purpose, we have chosen the DFTB semi-empirical method, which has been proven to provide accurate results at an affordable computational cost [33].

Before the MD simulations, all of the structures were optimized; the results of those calculations are presented in Table 10 and Table 11.

During the simulations, no significant conformational changes have been observed (Appendix A). Due to the experimentally determined very low stability of dehydrated **HH** and its almost instant amorphization, we have anticipated some changes either in the conformations of the molecules or in the unit cell dimensions. However, none of them have been observed; this could be possibly caused by a too-short simulation time, 20 ps, or an inadequate level of theory. However, during our previous studies [34,35], we have found that major changes accompanying polymorphic phase transitions occur during the first few picoseconds of simulations. However, in those mentioned cases, we have performed the MD simulations at the DFT level using CASTEP due to the significantly smaller unit cells in those previous studies.

## 3. Materials and Methods

### 3.1. Periodic DFT Calculations

The density functional theory (DFT) calculations of geometry optimization and NMR parameters under periodic boundary conditions were carried out with the CASTEP program [36] implemented in the Materials Studio 2020 software [37] using the plane wave pseudopotential formalism. Ultrasoft pseudopotentials were generated using the Koelling–Harmon scalar relativistic approach [38]. The Perdew–Burke–Ernzerhof (PBE) [39] exchange–correlation functional, defined within the generalized gradient approximation, with the Tkatchenko–Scheffler (TS) [40] dispersion correction, was used in the calculations.

#### 3.1.1. Geometry Optimization

Geometry optimization was carried out using the limited memory Broyden–Fletcher–Goldfarb–Shanno (LBFGS) [41] optimization scheme and smart method for finite basis set correction. The kinetic energy cutoff for the plane waves (E_cut_) was set to 630.0 eV. The number of Monkhorst–Pack k-points during sampling for a primitive cell Brillouin zone integration [42] were set to 2 × 1 × 1 (for thiamine monohydrate-based structures) and 1 × 2 × 1 (for thiamine hemihydrate based structures), respectively. The details on the structure preparation can be found in Section 2.1.

During geometry optimization, all atoms’ positions and the cell parameters were optimized with no constraints. The convergence criteria were set at 5 × 10^−6^ eV/atom for the energy, 1 × 10^−2^ eV/Å for the interatomic forces, 2 × 10^−2^ GPa for the stresses, and 5 × 10^−4^ Å for the maximum displacement. The fixed basis set quality method for the cell optimization calculations and the 5 × 10^−7^ eV/atom tolerance for SCF were used.

#### 3.1.2. NMR Parameters Calculations

The computation of shielding tensors was performed using the Gauge Including Projector Augmented Wave Density Functional Theory (GIPAW) method of Pickard et al. [43]. To compare the theoretical and experimental data, the calculated chemical shielding constants (σiso) were converted to chemical shifts (δiso) using the following equation: δiso = (σGly + δGly) − σiso, where σGly and δGly stand for the shielding constant and the experimental chemical shift, respectively, of the glycine carbonyl carbon atom (176.50 ppm).

### 3.2. Periodic DFTB Calculations

The Density Functional Tight Binding (DFTB) calculations of geometry optimization and molecular dynamics simulations under periodic boundary conditions were carried out with the DFTB+ program [44] implemented in the Materials Studio 2020 software [37]. The calculations utilized a library containing Slater–Koster atomic parameters, incorporating the UFF-based Lennard–Jones dispersion corrections and charge self-consistency.

#### 3.2.1. Geometry Optimization

Geometry optimization was carried out using the “smart” algorithm for calculations, the “divide and conquer” method for diagonalizing the Hamiltonian (eigensolver), the Broyden charge mixing scheme, and the Methfessel–Paxton distribution function used for smearing. The number of Monkhorst–Pack k-points during sampling for a primitive cell Brillouin zone integration [42] were set to 4 × 1 × 2 (for thiamine monohydrate-based structures) and 1 × 4 × 1 (for thiamine hemihydrate based structures), respectively. The details on the structure preparation can be found in Section 2.1.

During geometry optimization, all atoms’ positions and the cell parameters were optimized with no constraints. The convergence criteria were set at 1 × 10^−2^ kcal/mol for the energy, 5 × 10^−2^ kcal/mol/Å for the interatomic forces, 2 × 10^−2^ GPa for the stresses, and 5 × 10^−4^ Å for the maximum displacement and the 1 × 10^−8^ kcal/mol tolerance for SCC were used.

#### 3.2.2. Molecular Dynamics Simulations

Molecular dynamics (MD) simulations were run using an NPT ensemble maintained at a constant temperature of 293 K and pressure of 0.01 GPa, using a Nosé thermostat with 0.01 Q ratio and Berendsen barostat with 0.1 ps decay constant. The time step was set to 0.5 fs, and the total time of the simulation was set to 20 ps. All of the settings and electronic options were set at the same values as for geometry optimization (Section 3.2.1). No symmetry constraints were applied during the simulations.

## 4. Conclusions

In this work, two different forms of **THCL** hydrates, **NSH** and **HH**, have been studied using various molecular modeling methods. The first step of this work included the creation of the structures of hydrates with decreasing water: **THCL** molar ratio, starting either from monohydrate (**NSH**) or hemihydrate (**HH**) up to the fully dehydrated forms. After the optimization of all of the structures, we have found a good agreement between theoretically modeled and experimentally determined values describing the unit cell dimensions; this indicates that modeled structures obtained for the forms that have not been studied experimentally yet should also be accurate. Also, a comparison of the energy of the structures based on **NSH** and **HH** was in agreement with the experimental findings, indicating higher relative stability of the **NSH**.

GIPAW NMR calculations performed for the optimized structures allowed not only to assign all of the peaks in the experimental ^13^C CP MAS NMR spectra to particular carbon atoms but also to explain, at the molecular level, all of the changes observed in the spectra occurring as a result of the experimental dehydration. Comparison of the NMR calculations results with the energy of the partially dehydrated structures enables us to predict the structure of the intermediate forms occurring between the fully hydrated forms of **NSH** and **HH** and their dehydrated counterparts.

During the molecular dynamics simulations performed at the semi-empirical QM level, we did not observe major changes in the studied structures. For all of the forms of **NSH**, those results were anticipated, as this form is a variable hydrate that can exist at various THCL: water ratios, depending on the air humidity. However, the lack of changes in the unit cell dimensions of the dehydrated structure of **HH** is intriguing, as the previous experimental works reported on the instability of this form. We aim to investigate this aspect deeply in the near future.

This work highlights the accuracy and versatility of the “solid-state DFT” calculations in the analysis of various forms of molecular solids, in particular solvates of active pharmaceutical ingredients of various degrees of hydration.

## Figures and Tables

**Figure 1 molecules-28-07497-f001:**
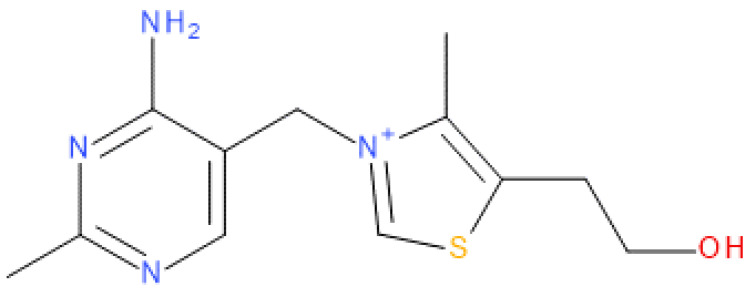
Chemical structure of thiamine cation.

**Figure 2 molecules-28-07497-f002:**
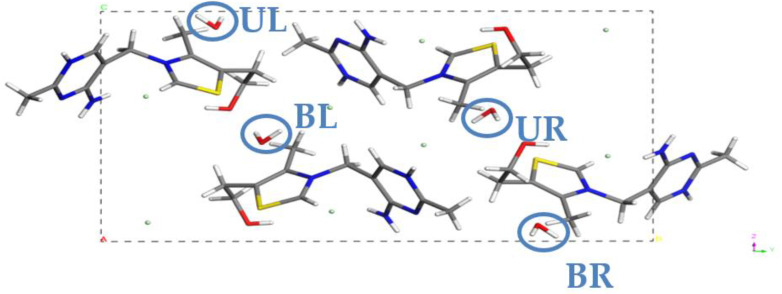
Unit cell of **NSH**, refcode **THIAMC12**. Water molecules have been named UL (**upper left**), BL (**bottom left**), UR (**upper right**), and BR (**bottom right**).

**Figure 3 molecules-28-07497-f003:**
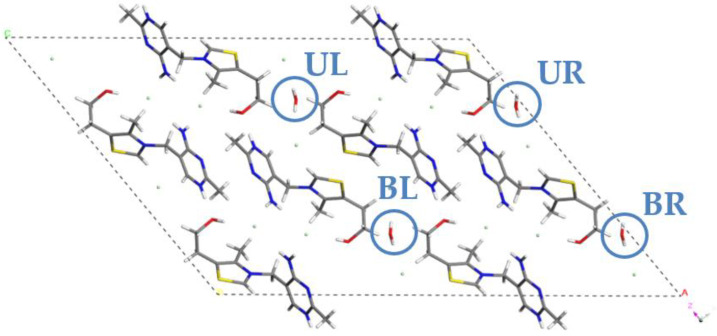
Unit cell of **HH**, refcode **WUWJAA**. Water molecules have been named UL (**upper left**), BL (**bottom left**), UR (**upper right**), and BR (**bottom right**).

**Figure 4 molecules-28-07497-f004:**
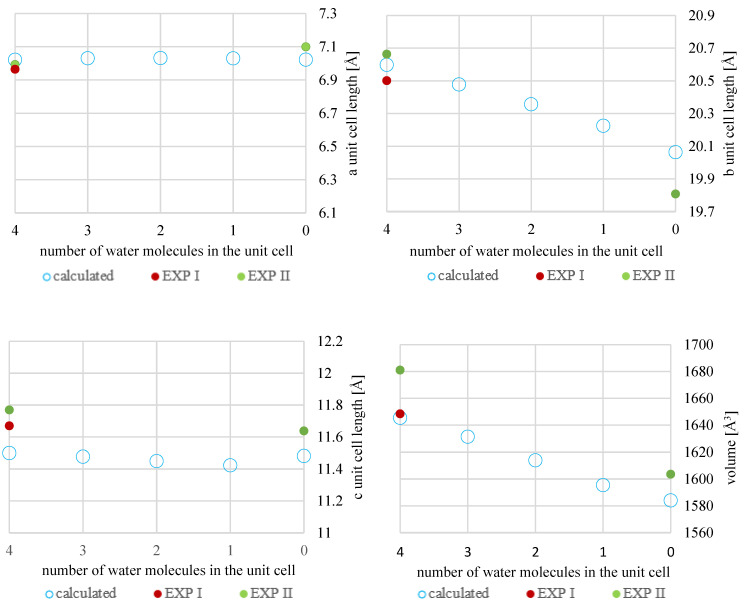
Experimental (exp I, exp II) and calculated unit cell dimensions of **MH**; for better comparison, the equal range of vertical axis–1.2 Å–has been used for all three unit cell lengths (a, b, and c). For **MH2W**, the arithmetic mean of the results (**MH2W BLUR, MH2W ULBL**, and **MH2W ULUR**) was shown.

**Figure 5 molecules-28-07497-f005:**
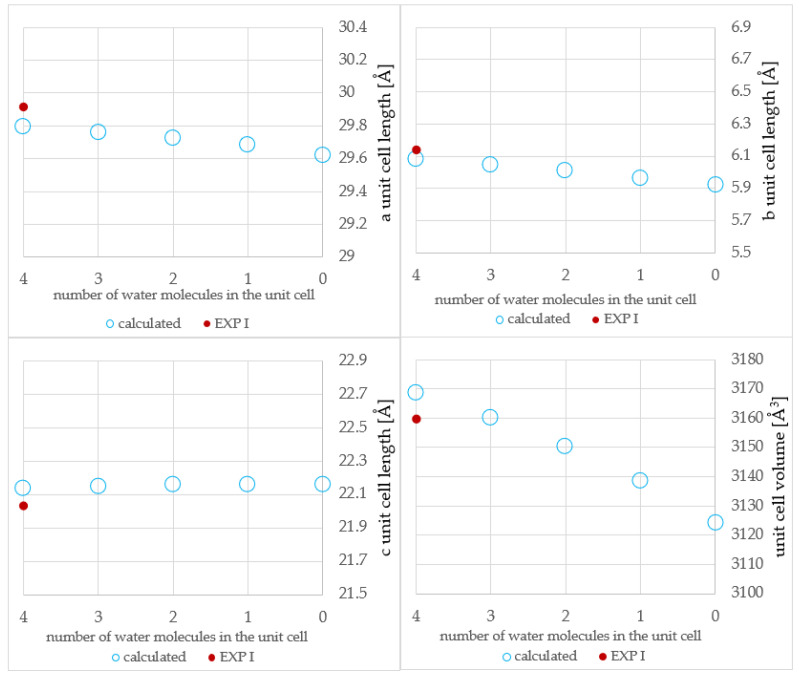
Experimental (exp I) and calculated unit cell dimensions of **HH**; for better comparison, the equal range of vertical axis–1.4 Å–has been used for all three unit cell lengths (a, b, and c). For **HH2W**, the arithmetic mean of the results (**HH2W BLUR**, **HH2W ULBL**, and **HH2W ULUR**) was shown.

**Figure 6 molecules-28-07497-f006:**
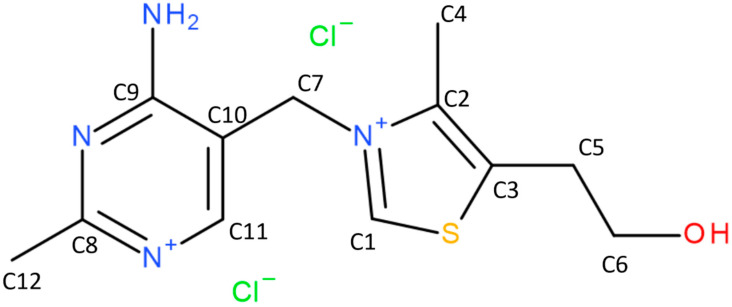
Carbon atom numbering of **THCL** used in the NMR analysis.

**Figure 7 molecules-28-07497-f007:**
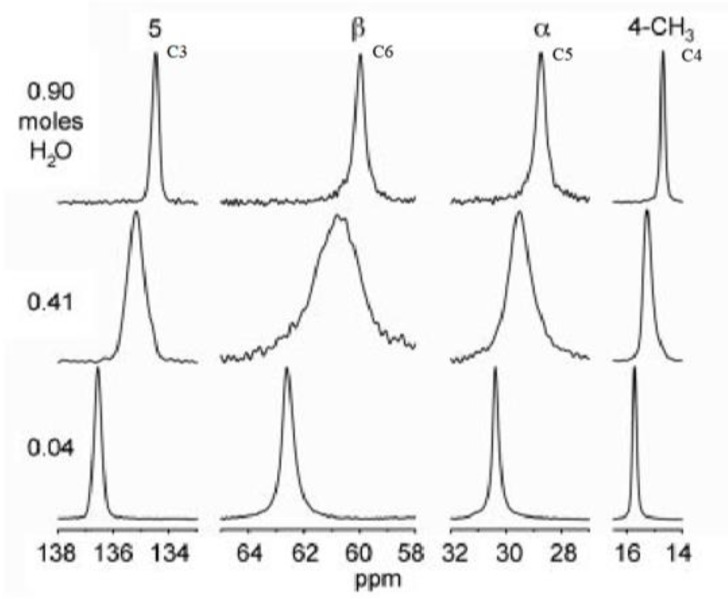
Chosen regions of the ^13^C CP MAS NMR spectra of NSH at various degrees of hydration (0.90, 0.41, and 0.04). Reprinted from [28] with permission from Elsevier.

**Figure 8 molecules-28-07497-f008:**
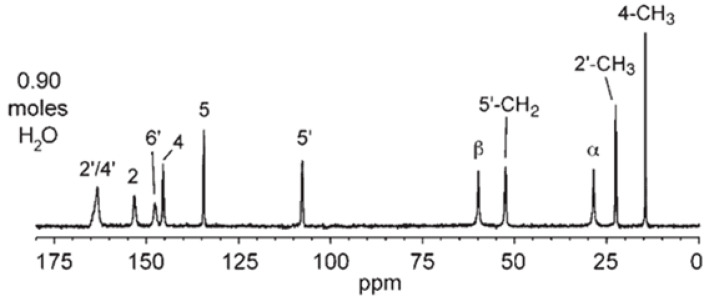
^13^C CP MAS NMR spectra of 0.9 hydrate of NSH. Reprinted from [28] with permission from Elsevier.

**Figure 9 molecules-28-07497-f009:**
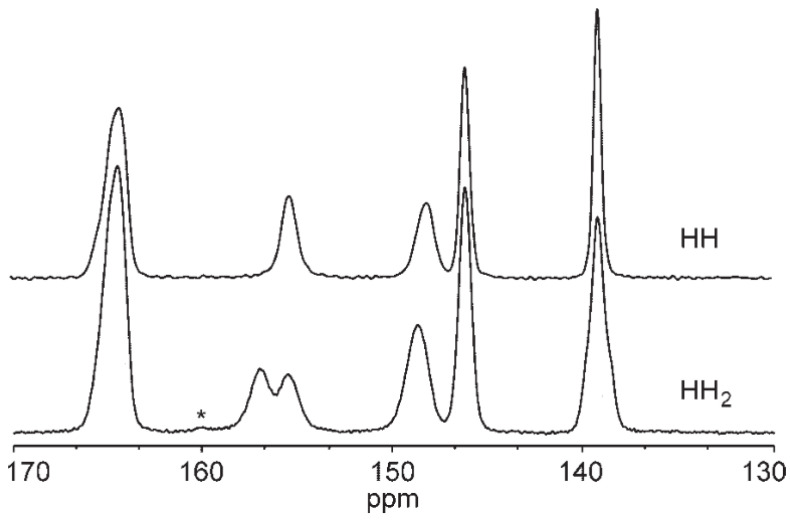
The chosen region of the ^13^C CP MAS NMR spectra of HH and the partially dehydrated form of **HH** named **HH_2_**. Asterisk indicates spinning sideband. Reprinted from [27] with permission from Elsevier.

**Figure 10 molecules-28-07497-f010:**
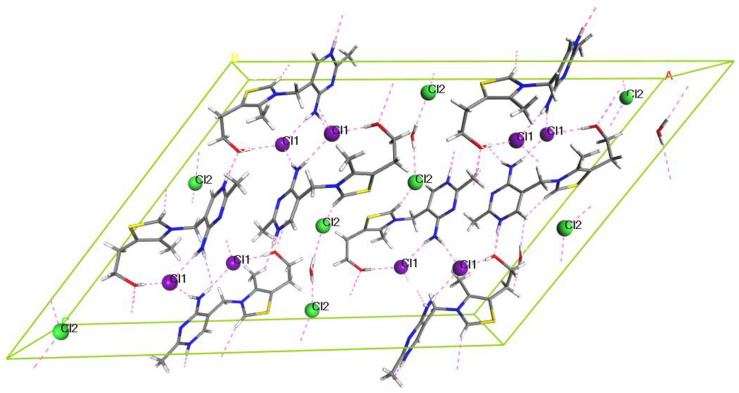
Optimized crystal unit cell of HH4W; pink dashed lines represent the intermolecular interactions. Atom coloring: N—blue, C grey, S—yellow, H—white, and O—red. The colors of symmetry equivalent Cl atoms are either violet (Cl1) or green (Cl2).

**Figure 11 molecules-28-07497-f011:**
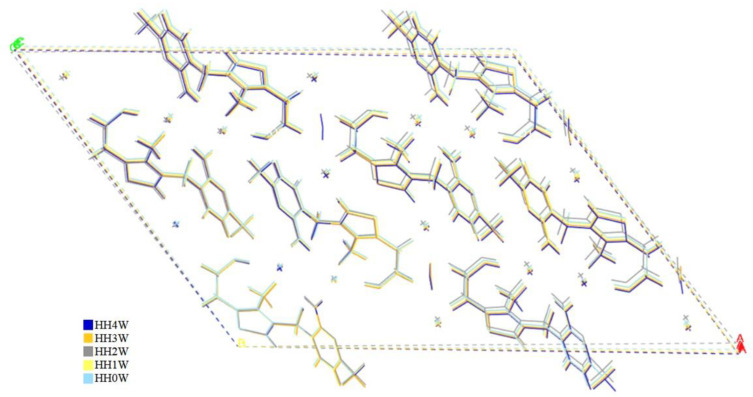
Optimized crystal unit cells of HH4W, HH3W, HH2W, HH1W, and HH0W. For comparison, please refer to Figure 10.

**Table 1 molecules-28-07497-t001:** Crystallographic information concerning the chosen structures of thiamine hydrochloride.

CCDC Refcode	Space Group	Z	T [K]	a [Å]	b [Å]	c [Å]	α [°]	β [°]	γ [°]	V [Å^3^]	Structure Description	Structure Code in This Work	Ref.
**THIAMC12**	*P*2_1_/*n*	4	296	6.993	20.663	11.77	90	98.699	90	1681	Thiamine hydrochloride monohydrate	**EXP II MH4W**	[29]
**THIAMC14**	*P*2_1_/*n*	4	173	6.965	20.501	11.67	90	98.406	90	1648	Thiamine hydrochloride monohydrate	**EXP I MH4W**	[27]
**UNEXOA**	*P*2_1_/*n*	4	263	7.099	19.808	11.638	90	101.529	90	1604	Thiamine hydrochloride anhydrous	**EXP II MH0W**	[29]
**WUWJAA**	*C*2/*c*	8	173	29.921	6.138	22.041	90	128.684	90	3160	Thiamine hydrochloride hemihydrate	**EXP I** **HH4W**	[27]

**Table 2 molecules-28-07497-t002:** Structures used for calculations prepared from the **THIAMC12-NSH**.

Code	Water Molecules Removed	Water Molecules Left	Water: THCL Stoichiometry
**MH4W**	none	UL, BL, UR, BR	1
**MH3W**	UL	BL, UR, RR	0.75
**MH2W BLUR**	UL, BR	BL, UR	0.5
**MH2W ULBL**	UR, BR	UL, BL	0.5
**MH2W ULUR**	BL, BR	UL, UR	0.5
**MH1W**	BL, UR, RR	UL	0.25
**MH0W**	UL, BL, UR, BR	none	0

**Table 3 molecules-28-07497-t003:** Structures used for calculations prepared from the **WUWJAA-HH**.

Code	Water Molecules Removed	Water Molecules Left	Water: THCL Stoichiometry
**HH4W**	none	UL, BL, UR, BR	0.5
**HH3W**	UL	BL, UR, RR	0.375
**HH2W BLUR**	UL, BR	BL, UR	0.25
**HH2W ULBL**	UR, BR	UL, BL	0.25
**HH2W ULUR**	BL, BR	UL, UR	0.25
**HH1W**	BL, UR, RR	UL	0.125
**HH0W**	UL, BL, UR, BR	none	0

**Table 4 molecules-28-07497-t004:** Optimized unit cell dimensions of the **MH** structures of various hydration ratios, compared with the experimental ones.

Code	Energy [kcal/mol]	Relative Energy [kcal/mol]	a [Å]	b [Å]	c [Å]	α [°]	β [°]	γ [°]	V [Å^3^]
**MH4W**	−499,908.604	−43,190.805	7.022	20.598	11.500	89.998	98.368	90.012	1645.507
**MH3W**	−489,110.139	−32,392.340	7.031	20.478	11.477	90.031	99.148	90.647	1631.380
**MH2W BLUR**	−478,312.186	−21,594.387	7.034	20.365	11.462	89.858	100.184	91.472	1615.564
**MH2W ULBL**	−478,312.416	−21,594.617	7.021	20.364	11.449	90.003	99.778	90.006	1613.213
**MH2W ULUR**	−478,311.772	−21,593.973	7.031	20.337	11.424	90.004	99.974	89.984	1608.722
**MH1W**	−467,514.597	−10,796.798	7.030	20.225	11.423	89.945	100.793	90.740	1595.441
**MH0W**	−456,717.799	0	7.022	20.064	11.482	90.004	101.714	90.002	1584.016
**EXPII MH4W**	-		6.993	20.663	11.770	90.000	98.699	90.000	1681.047
**EXPII MH0W**	-		7.099	19.808	11.638	90.000	101.529	90.000	1603.545
**EXPI MH4W**	-		6.965	20.501	11.670	90.000	98.406	90.000	1648.428

**Table 5 molecules-28-07497-t005:** Optimized unit cell dimensions of the **HH** structures of various hydration ratios, compared with the experimental ones.

Code	Energy [kcal/mol]	Relative Energy [kcal/mol]	a [Å]	b [Å]	c [Å]	α [°]	β [°]	γ [°]	V [Å^3^]
**HH4W**	−956,630.306	−43,198.085	29.798	6.083	22.141	90.089	127.855	89.930	3168.616
**HH3W**	−945,830.348	−32,398.127	29.761	6.046	22.150	90.030	127.541	89.978	3160.184
**HH2W BLUR**	−935,030.608	−21,598.387	29.734	6.009	22.160	90.005	127.260	89.995	3151.170
**HH2W ULBL**	−935,030.916	−21,598.695	29.726	6.003	22.162	90.105	127.230	89.915	3148.795
**HH2W ULUR**	−935,030.539	−21,598.318	29.730	6.012	22.161	89.996	127.289	90.001	3151.079
**HH1W**	−924,231.287	−10,799.066	29.686	5.966	22.167	89.961	126.924	90.031	3138.81
**HH0W**	−913,432.221	0	29.618	5.922	22.164	90.000	126.522	90.000	3124.258
**EXP HH4W**	-		29.921	6.138	22.041	90.000	128.684	90.000	3160.049

**Table 6 molecules-28-07497-t006:** Energy values of the optimized structures. Due to the differences in Z, eight for HH, and four for MH, values obtained for HH have been divided by two to enable the comparison. Explanations of the abbreviations can be found in Table 2 and Table 3.

Code	Energy [kcal/mol]	Code	Energy [kcal/mol]	Δ(HH-MH) [kcal/mol]
THCL: water ratio 2:1				
**HH4W**	−478,315.153	**MH2W BLUR**	−478,312.186	−2.967
		**MH2W ULBL**	−478,312.416	−2.737
		**MH2W ULUR**	−478,311.772	−3.381
THCL: water ratio 4:1				
**HH2W BLUR**	−467,515.304	**MH1W**	−467,514.597	−0.707
**HH2W ULBL**	−467,515.458			−0.861
**HH2W ULUR**	−467,515.270			−0.672
Anhydrous structures				
**HH0W**	−456,716.111	**MH0W**	−456,717.799	1.688

**Table 7 molecules-28-07497-t007:** Dehydration energies are defined as the energy required to remove one water molecule for each step (1–4) of this process. HH—hemihydrate, MH—monohydrate.

Step	Number of Water Molecules	HH [kcal/mol]	MH [kcal/mol]	Δ(HH-MH) [kcal/mol]
**1**	4 W → 3 W	18.848	17.356	1.492
**2**	3 W → 2 W	18.322	16.613	1.709
**3**	2 W → 1 W	18.520	16.709	1.811
**4**	1 W → 0 W	17.956	15.688	2.268

**Table 8 molecules-28-07497-t008:** Calculated chemical shift values for carbon atoms, compared with the experimental ones. Atom numbering is presented in Figure 6. Explanations of the abbreviations (structure codes) can be found in Table 2. Due to Z = 4, for each carbon atom, four theoretical values have been obtained from calculations. To facilitate the analysis of the data in the table, the two-color scale was applied. The cell that holds the minimum is colored red, and the cell that holds the maximum value is colored blue. All other cells are colored proportionally. SD—standard deviation of the four values.

Structure Code	C8					C9					C1					C11					C2					C3				
	SD	1 [ppm]	2 [ppm]	3 [ppm]	4 [ppm]	SD	1 [ppm]	2 [ppm]	3 [ppm]	4 [ppm]	SD	1 [ppm]	2 [ppm]	3 [ppm]	4 [ppm]	SD	1 [ppm]	2 [ppm]	3 [ppm]	4 [ppm]	SD	1 [ppm]	2 [ppm]	3 [ppm]	4 [ppm]	SD	1 [ppm]	2 [ppm]	3 [ppm]	4 [ppm]
**MH4W**	0.01	163.92	163.95	163.94	163.95	0.02	160.45	160.43	160.46	160.41	0.01	149.78	149.79	149.76	149.78	0.01	148.00	148.00	148.02	148.00	0.02	147.21	147.19	147.22	147.18	0.02	140.73	140.77	140.74	140.79
**MH3W**	0.83	162.54	163.93	163.91	164.87	0.12	160.57	160.42	160.50	160.26	0.43	149.40	148.45	149.23	149.56	0.23	147.99	148.35	148.30	147.79	0.79	147.23	148.65	146.52	147.02	1.24	140.93	139.83	143.26	141.22
**MH2W BLUR**	1.28	162.52	165.07	162.54	165.09	0.22	160.75	160.30	160.74	160.30	0.35	149.05	148.33	149.04	148.36	0.12	148.26	148.02	148.25	148.02	0.95	146.61	148.52	146.59	148.49	1.75	143.61	140.10	143.60	140.11
**MH2W ULBL**	0.27	163.94	163.40	163.40	163.94	0.02	160.51	160.47	160.47	160.51	0.56	148.35	149.46	149.47	148.33	0.27	148.44	147.91	147.90	148.45	0.32	147.70	147.09	147.05	147.72	0.53	142.59	141.44	141.52	142.50
**MH2W ULUR**	1.21	164.89	164.88	162.46	162.46	0.14	160.31	160.32	160.59	160.59	0.59	149.11	149.15	147.93	147.96	0.10	147.96	147.97	148.18	148.16	1.16	146.38	146.33	148.68	148.67	1.96	143.77	143.78	139.87	139.84
**MH1W**	0.96	162.39	163.48	163.49	165.07	0.17	160.78	160.45	160.59	160.31	0.41	147.95	148.15	149.04	148.28	0.10	148.33	148.07	148.10	148.11	0.75	147.72	148.54	146.43	147.71	1.49	142.84	140.04	144.15	142.71
**MH0W**	0.01	162.68	162.68	162.69	162.67	0.01	160.96	160.94	160.95	160.95	0.01	148.41	148.41	148.38	148.42	0.00	148.27	148.27	148.28	148.27	0.01	148.29	148.28	148.28	148.27	0.01	143.28	143.27	143.26	143.28
**EXPII MH4W**	0.00	163.50	163.50	163.50	163.50	0.00	163.50	163.50	163.50	163.50	0.00	153.20	153.20	153.20	153.20	0.00	147.70	147.70	147.70	147.70	0.00	145.50	145.50	145.50	145.50	0.00	134.50	134.50	134.50	134.50
**EXPII MH0W**	0.00	163.20	163.20	163.20	163.20	0.00	163.20	163.20	163.20	163.20	0.00	153.10	153.10	153.10	153.10	0.00	148.00	148.00	148.00	148.00	0.00	145.50	145.50	145.50	145.50	0.00	136.40	136.40	136.40	136.40
**EXPI 0.90H_2_O**	0.00	163.20	163.20	163.20	163.20	0.00	163.20	163.20	163.20	163.20	0.00	153.10	153.10	153.10	153.10	0.00	147.70	147.70	147.70	147.70	0.00	145.20	145.20	145.20	145.20	0.00	134.50	134.50	134.50	134.50
**EXPI 0.41H_2_O**	0.00	163.00	163.00	163.00	163.00	0.00	163.00	163.00	163.00	163.00	0.00	153.00	153.00	153.00	153.00	0.00	147.70	147.70	147.70	147.70	0.00	145.20	145.20	145.20	145.20	0.00	135.20	135.20	135.20	135.20
**EXPI 0.04H_2_O**	0.00	163.00	163.00	163.00	163.00	0.00	163.00	163.00	163.00	163.00	0.00	153.00	153.00	153.00	153.00	0.00	147.70	147.70	147.70	147.70	0.00	145.20	145.20	145.20	145.20	0.00	136.50	136.50	136.50	136.50
**MH0W-MH4W**		−1.24	−1.27	−1.25	−1.28		0.51	0.51	0.49	0.54		−1.37	−1.38	−1.38	−1.36		0.27	0.27	0.26	0.27		1.08	1.09	1.06	1.09		2.55	2.50	2.52	2.49
**EXPII MH0W-EXPII MH4W**		−0.30	−0.30	−0.30	−0.30		−0.30	−0.30	−0.30	−0.30		−0.10	−0.10	−0.10	−0.10		0.30	0.30	0.30	0.30		0.00	0.00	0.00	0.00		1.90	1.90	1.90	1.90
**Structure Code**	**C10**					**C6**					**C7**					**C5**					**C12**					**C4**				
	**SD**	**1 [ppm]**	**2 [ppm]**	**3 [ppm]**	**4 [ppm]**	**SD**	**1 [ppm]**	**2 [ppm]**	**3 [ppm]**	**4 [ppm]**	**SD**	**1 [ppm]**	**2 [ppm]**	**3 [ppm]**	**4 [ppm]**	**SD**	**1 [ppm]**	**2 [ppm]**	**3 [ppm]**	**4 [ppm]**	**SD**	**1 [ppm]**	**2 [ppm]**	**3 [ppm]**	**4 [ppm]**	**SD**	**1 [ppm]**	**2 [ppm]**	**3 [ppm]**	**4 [ppm]**
**MH4W**	0.01	109.25	109.27	109.25	109.26	0.00	62.26	62.26	62.26	62.25	0.01	51.47	51.50	51.48	51.47	0.01	26.58	26.60	26.59	26.58	0.02	19.42	19.47	19.45	19.45	0.01	12.47	12.47	12.45	12.48
**MH3W**	0.45	109.96	108.94	108.83	109.07	0.21	62.27	62.13	61.8	62.35	0.44	51.22	52.23	52.11	51.40	0.33	26.62	27.43	27.26	26.79	0.49	20.50	19.57	19.16	19.57	0.62	12.12	13.60	12.33	12.08
**MH2W BLUR**	0.51	109.68	108.65	109.66	108.64	0.31	61.76	62.37	61.75	62.37	0.15	51.98	52.27	51.99	52.28	0.22	27.32	27.74	27.31	27.78	0.19	20.25	19.83	20.18	19.83	0.67	12.00	13.40	12.03	13.32
**MH2W ULBL**	0.59	108.68	109.86	109.85	108.68	0.29	61.85	62.44	62.43	61.87	0.80	52.91	51.30	51.30	52.87	0.47	27.88	26.94	26.95	27.87	0.55	19.43	20.52	20.53	19.43	0.95	13.69	11.84	11.81	13.74
**MH2W ULUR**	0.61	108.55	108.54	109.77	109.75	0.21	61.83	61.82	62.23	62.25	0.02	51.99	51.98	52.03	52.00	0.05	27.30	27.34	27.42	27.42	0.67	19.46	19.42	20.79	20.75	0.65	12.10	12.10	13.40	13.41
**MH1W**	0.50	109.40	109.32	109.30	108.20	0.23	61.79	62.41	61.91	62.01	0.37	52.78	52.10	51.96	52.75	0.22	27.88	27.90	27.44	28.03	0.43	20.52	20.75	20.39	19.60	0.76	13.42	13.06	11.73	13.72
**MH0W**	0.01	108.47	108.45	108.47	108.46	0.01	64.09	64.1	64.09	64.11	0.00	52.86	52.86	52.86	52.86	0.01	27.96	27.95	27.95	27.97	0.01	21.03	21.04	21.04	21.02	0.02	13.20	13.24	13.21	13.23
**EXPII MH4W**	0.00	107.70	107.70	107.70	107.70	0.00	59.90	59.90	59.90	59.90	0.00	52.40	52.40	52.40	52.40	0.00	28.30	28.30	28.30	28.30	0.00	22.60	22.60	22.60	22.60	0.00	14.60	14.60	14.60	14.60
**EXPII MH0W**	0.00	107.10	107.10	107.10	107.10	0.00	62.40	62.40	62.40	62.40	0.00	52.60	52.60	52.60	52.60	0.00	30.10	30.10	30.10	30.10	0.00	23.10	23.10	23.10	23.10	0.00	15.60	15.60	15.60	15.60
**EXPI 0.90H_2_O**	0.00	107.80	107.80	107.80	107.80	0.00	60.00	60.00	60.00	60.00	0.00	52.50	52.50	52.50	52.50	0.00	28.70	28.70	28.70	28.70	0.00	22.50	22.50	22.50	22.50	0.00	14.70	14.70	14.70	14.70
**EXPI 0.41H_2_O**	0.00	107.50	107.50	107.50	107.50	0.00	60.80	60.80	60.80	60.80	0.00	52.60	52.60	52.60	52.60	0.00	29.50	29.50	29.50	29.50	0.00	22.60	22.60	22.60	22.60	0.00	15.30	15.30	15.30	15.30
**EXPI 0.04H_2_O**	0.00	107.00	107.00	107.00	107.00	0.00	62.60	62.60	62.60	62.60	0.00	52.70	52.70	52.70	52.70	0.00	30.40	30.40	30.40	30.40	0.00	23.00	23.00	23.00	23.00	0.00	15.70	15.70	15.70	15.70
**MH0W-MH4W**		−0.78	−0.82	−0.78	−0.80		1.83	1.84	1.83	1.86		1.39	1.36	1.38	1.39		1.38	1.35	1.36	1.39		1.61	1.57	1.59	1.57		0.73	0.77	0.76	0.75
**EXPII MH0W-EXPII MH4W**		−0.60	−0.60	−0.60	−0.60		2.50	2.50	2.50	2.50		0.20	0.20	0.20	0.20		1.80	1.80	1.80	1.80		0.50	0.50	0.50	0.50		1.00	1.00	1.00	1.00

**Table 9 molecules-28-07497-t009:** Calculated chemical shift values for carbon atoms, compared with the experimental ones. Atom numbering is presented in Figure 6. Explanations of the abbreviations (structure codes) can be found in Table 2. Due to Z = 8, for each carbon atom, eight theoretical values have been obtained from calculations. To facilitate the analysis of the data in the table, the two-color scale was applied. The cell that holds the minimum is colored red, and the cell that holds the maximum value is colored blue. All other cells are colored proportionally. SD—standard deviation of the four values.

Structure Code	C8									C9									C1								
	SD	1 [ppm]	2 [ppm]	3 [ppm]	4 [ppm]	5 [ppm]	6 [ppm]	7 [ppm]	8 [ppm]	SD	1 [ppm]	2 [ppm]	3 [ppm]	4 [ppm]	5 [ppm]	6 [ppm]	7 [ppm]	8 [ppm]	SD	1 [ppm]	2 [ppm]	3 [ppm]	4 [ppm]	5 [ppm]	6 [ppm]	7 [ppm]	8 [ppm]
**HH4W**	0.04	164.49	164.39	164.42	164.50	164.45	164.44	164.41	164.47	0.03	164.73	164.65	164.66	164.72	164.69	164.66	164.71	164.73	0.06	155.6	155.57	155.73	155.72	155.65	155.67	155.72	155.6
**HH3W**	0.17	164.66	164.49	164.64	164.57	164.31	164.24	164.29	164.26	0.05	164.74	164.71	164.71	164.75	164.71	164.59	164.72	164.74	0.44	155.81	155.63	155.85	155.73	155.75	156.77	155.73	156.81
**HH2W ULBL**	0.07	164.37	164.44	164.30	164.48	164.35	164.40	164.29	164.46	0.09	164.59	164.67	164.54	164.76	164.56	164.66	164.53	164.59	0.45	156.66	155.75	156.77	155.88	156.67	155.76	156.76	156.66
**HH2W BLUR**	0.04	164.43	164.30	164.41	164.36	164.35	164.41	164.36	164.43	0.04	164.65	164.69	164.61	164.71	164.71	164.6	164.7	164.65	0.52	156.81	155.77	156.86	155.81	155.8	156.86	155.78	156.81
**HH2W ULUR**	0.30	164.12	163.97	164.00	164.09	164.62	164.64	164.64	164.63	0.06	164.62	164.54	164.54	164.63	164.69	164.64	164.64	164.62	0.46	156.76	156.67	156.71	156.7	155.82	155.75	155.75	156.76
**HH1W**	0.16	164.23	164.05	164.15	164.11	164.45	164.45	164.43	164.43	0.07	164.58	164.49	164.52	164.51	164.68	164.47	164.63	164.58	0.42	156.81	156.72	156.74	156.7	155.87	156.84	155.75	156.81
**HH0W**	0.01	164.29	164.31	164.30	164.30	164.30	164.28	164.30	164.29	0.01	164.52	164.52	164.52	164.51	164.51	164.51	164.51	164.52	0.02	156.79	156.81	156.75	156.81	156.78	156.8	156.77	156.79
**EXP HH4W**	0.00	164.50	164.50	164.50	164.50	164.50	164.50	164.50	164.50	0.00	164.50	164.50	164.50	164.50	164.50	164.50	164.50	164.50	0.00	155.50	155.50	155.50	155.50	155.50	155.50	155.50	155.50
HH0W-HH4W		−0.20	−0.08	−0.12	−0.20	−0.15	−0.16	−0.11	−0.18		−0.21	−0.13	−0.14	−0.21	−0.18	−0.15	−0.20	−0.24		**1.19**	**1.24**	**1.02**	**1.09**	**1.13**	**1.13**	**1.05**	**1.07**
**Structure Code**	**C11**									**C2**									**C3**								
	**SD**	**1** **[ppm]**	**2** **[ppm]**	**3** **[ppm]**	**4** **[ppm]**	**5** **[ppm]**	**6** **[ppm]**	**7** **[ppm]**	**8** **[ppm]**	**SD**	**1** **[ppm]**	**2** **[ppm]**	**3** **[ppm]**	**4** **[ppm]**	**5** **[ppm]**	**6** **[ppm]**	**7** **[ppm]**	**8** **[ppm]**	**SD**	**1** **[ppm]**	**2** **[ppm]**	**3** **[ppm]**	**4** **[ppm]**	**5** **[ppm]**	**6** **[ppm]**	**7** **[ppm]**	**8** **[ppm]**
**HH4W**	0.02	148.87	148.85	148.88	148.92	148.90	148.87	148.89	148.91	0.06	147.45	147.37	147.35	147.43	147.33	147.33	147.47	147.48	0.25	140.16	140.62	140.56	140.12	140.62	140.62	140.06	140.16
**HH3W**	0.18	149.04	149.17	149.04	149.21	148.93	148.68	148.91	148.72	0.10	147.37	147.53	147.28	147.48	147.44	147.24	147.51	147.34	0.37	140.43	139.68	140.73	139.68	140.6	140.51	140.19	140.43
**HH2W ULBL**	0.02	149.05	149.04	149.05	149.08	149.04	149.05	149.03	149.08	0.03	147.34	147.36	147.39	147.40	147.38	147.32	147.38	147.42	0.50	139.67	140.83	139.6	140.36	139.66	140.81	139.63	139.67
**HH2W BLUR**	0.19	148.84	149.23	148.87	149.23	149.24	148.83	149.21	148.87	0.19	147.23	147.58	147.13	147.56	147.55	147.18	147.60	147.24	0.48	140.42	139.64	140.75	139.62	139.66	140.71	139.59	140.42
**HH2W ULUR**	0.29	148.74	148.68	148.72	148.69	149.31	149.26	149.28	149.30	0.06	147.31	147.27	147.26	147.32	147.40	147.40	147.43	147.38	0.33	140.13	140.6	140.55	140.16	139.81	139.77	139.79	140.13
**HH1W**	0.18	148.86	149.04	148.83	149.07	149.36	149.18	149.32	149.22	0.10	147.26	147.43	147.19	147.42	147.47	147.28	147.49	147.32	0.38	140.32	139.56	140.72	139.6	139.79	139.72	139.8	140.32
**HH0W**	0.01	149.29	149.31	149.32	149.30	149.29	149.30	149.30	149.31	0.01	147.38	147.39	147.41	147.39	147.39	147.39	147.40	147.41	0.01	139.81	139.77	139.8	139.79	139.81	139.8	139.81	139.81
**EXP HH4W**	0.00	148.00	148.00	148.00	148.00	148.00	148.00	148.00	148.00	0.00	146.00	146.00	146.00	146.00	146.00	146.00	146.00	146.00	0.00	140.00	140.00	140.00	140.00	140.00	140.00	140.00	140.00
**HH0W-HH4W**		0.42	0.46	0.44	0.38	0.39	0.43	0.41	0.40		−0.07	0.02	0.06	−0.04	0.06	0.06	−0.07	−0.07		−0.35	−0.85	−0.76	−0.33	−0.81	−0.82	−0.25	−0.28
**Structure Code**	**C10**									**C6**									**C7**								
	**SD**	**1 ** **[ppm]**	**2** **[ppm]**	**3 ** **[ppm]**	**4 ** **[ppm]**	**5 ** **[ppm]**	**6 ** **[ppm]**	**7 ** **[ppm]**	**8 ** **[ppm]**	SD	**1 ** **[ppm]**	**2 ** **[ppm]**	**3 ** **[ppm]**	**4 ** **[ppm]**	**5 ** **[ppm]**	**6 ** **[ppm]**	**7 ** **[ppm]**	**8 ** **[ppm]**	SD	**1 ** **[ppm]**	**2** **[ppm]**	**3 ** **[ppm]**	**4 ** **[ppm]**	**5 ** **[ppm]**	**6 ** **[ppm]**	**7 ** **[ppm]**	**8 ** **[ppm]**
**HH4W**	0.03	108.17	108.20	108.15	108.09	108.14	108.16	108.18	108.14	0.21	67.67	68.07	68.05	67.66	68.07	68.10	67.65	67.63	0.03	50.24	50.31	50.31	50.22	50.26	50.28	50.29	50.28
**HH3W**	0.22	108.15	108.16	108.12	108.09	108.24	108.68	108.29	108.67	0.22	67.82	67.48	68.15	67.54	68.00	67.97	67.64	67.75	0.06	50.11	50.17	50.17	50.15	50.25	50.07	50.26	50.13
**HH2W ULBL**	0.18	108.52	108.22	108.55	108.17	108.51	108.20	108.57	108.13	0.30	67.39	68.10	67.33	67.69	67.40	68.10	67.33	67.69	0.06	49.98	50.13	50.04	50.07	49.96	50.11	50.00	50.07
**HH2W BLUR**	0.23	108.61	108.18	108.62	108.13	108.14	108.65	108.21	108.63	0.25	67.73	67.42	68.04	67.45	67.49	68.01	67.39	67.78	0.09	49.95	50.16	49.98	50.11	50.11	49.96	50.14	49.96
**HH2W ULUR**	0.34	108.67	108.71	108.69	108.67	107.98	108.02	108.01	107.99	0.16	67.55	67.89	67.90	67.51	67.59	67.47	67.48	67.56	0.04	50.03	50.06	50.08	50.00	49.99	49.98	49.98	49.99
**HH1W**	0.22	108.59	108.58	108.59	108.54	108.04	108.39	108.05	108.40	0.20	67.58	67.24	67.92	67.27	67.48	67.39	67.40	67.43	0.04	49.87	49.93	49.88	49.91	49.94	49.82	49.94	49.83
**HH0W**	0.01	108.47	108.48	108.47	108.47	108.48	108.49	108.48	108.49	0.01	67.35	67.35	67.34	67.36	67.35	67.35	67.32	67.36	0.01	49.82	49.83	49.82	49.81	49.81	49.83	49.81	49.82
**EXP HH4W**	0.00	107.00	107.00	107.00	107.00	107.00	107.00	107.00	107.00	0.00	65.50	65.50	65.50	65.50	65.50	65.50	65.50	65.50	0.00	52.00	52.00	52.00	52.00	52.00	52.00	52.00	52.00
**HH0W-HH4W**		0.30	0.28	0.32	0.38	0.34	0.33	0.30	0.35		−0.32	−0.72	−0.71	−0.30	−0.72	−0.75	−0.33	−0.27		−0.42	−0.48	−0.49	−0.41	−0.45	−0.45	−0.48	−0.46
**Structure Code**	**C5**									**C12**									**C4**								
	**SD**	**1** **[ppm]**	**2** **[ppm]**	**3** **[ppm]**	**4** **[ppm]**	**5** **[ppm]**	**6** **[ppm]**	**7** **[ppm]**	**8** **[ppm]**	**SD**	**1** **[ppm]**	**2** **[ppm]**	**3** **[ppm]**	**4** **[ppm]**	**5** **[ppm]**	**6** **[ppm]**	**7** **[ppm]**	**8** **[ppm]**	**SD**	**1** **[ppm]**	**2** **[ppm]**	**3** **[ppm]**	**4** **[ppm]**	**5** **[ppm]**	**6** **[ppm]**	**7** **[ppm]**	**8** **[ppm]**
**HH4W**	0.14	30.22	29.96	29.96	30.22	29.93	29.94	30.23	30.21	0.05	24.43	24.44	24.52	24.53	24.49	24.42	24.49	24.43	0.11	15.66	15.52	15.56	15.71	15.59	15.41	15.62	15.78
**HH3W**	0.35	30.37	29.43	30.10	29.43	30.05	30.04	30.28	30.30	0.09	24.56	24.33	24.6	24.4	24.38	24.47	24.35	24.56	0.10	15.69	15.89	15.59	15.86	15.72	15.60	15.81	15.75
**HH2W ULBL**	0.40	29.50	30.10	29.53	30.44	29.52	30.10	29.53	30.45	0.06	24.28	24.34	24.29	24.45	24.29	24.32	24.32	24.28	0.13	15.96	15.62	15.91	15.90	15.93	15.63	15.91	15.89
**HH2W BLUR**	0.39	30.37	29.49	30.12	29.47	29.50	30.10	29.49	30.39	0.16	24.54	24.16	24.48	24.23	24.2	24.51	24.19	24.54	0.15	15.79	15.95	15.60	15.99	15.93	15.60	15.95	15.71
**HH2W ULUR**	0.37	30.36	29.99	29.98	30.35	29.47	29.49	29.50	29.44	0.06	24.27	24.23	24.24	24.26	24.38	24.34	24.33	24.27	0.07	15.82	15.68	15.67	15.85	15.83	15.78	15.76	15.89
**HH1W**	0.32	30.49	29.57	30.09	29.59	29.58	29.62	29.57	29.60	0.09	24.33	24.11	24.3	24.1	24.24	24.31	24.17	24.33	0.08	15.82	15.97	15.70	15.94	15.91	15.80	15.91	15.84
**HH0W**	0.01	29.72	29.75	29.73	29.76	29.73	29.75	29.74	29.76	0.01	24.23	24.26	24.23	24.24	24.26	24.25	24.26	24.23	0.01	16.00	15.98	16.01	15.98	15.98	15.97	15.98	15.98
**EXP HH4W**	0.00	31.00	31.00	31.00	31.00	31.00	31.00	31.00	31.00	0.00	24.00	24.00	24.00	24.00	24.00	24.00	24.00	24.00	0.00	18.50	18.50	18.50	18.50	18.50	18.50	18.50	18.50
**HH0W-HH4W**		−0.50	−0.21	−0.23	−0.46	−0.20	−0.19	−0.49	−0.45		−0.20	−0.18	−0.29	−0.29	−0.23	−0.17	−0.23	−0.31		0.34	0.46	0.45	0.27	0.39	0.56	0.36	0.20

**Table 10 molecules-28-07497-t010:** Optimized (DFTB+) unit cell dimensions of the **MH** structures of various hydration ratios, compared with the experimental ones.

Code	Energy [kcal/mol]	a [Å]	b [Å]	c [Å]	α [°]	β [°]	γ [°]	V [Å^3^]
**MH4W**	−133,137.10	7.014	20.831	11.139	90.000	99.644	90.000	1604.557
**MH3W**	−130,565.98	6.965	20.809	11.192	89.843	100.177	90.944	1596.371
**MH2W BLUR**	−127,995.21	6.905	20.796	11.272	90.07	101.034	91.423	1588.170
**MH2W ULBL**	−127,995.75	6.927	20.773	11.198	90.000	100.536	90.000	1584.090
**MH2W ULUR**	−127,994.89	6.967	20.497	11.294	90.000	100.753	90.000	1584.492
**MH1W**	−125,424.93	6.941	20.388	11.321	90.296	101.284	90.072	1571.063
**MH0W**	−122,856.04	6.934	20.078	11.402	90.000	101.664	90.000	1554.529
**EXPII MH4W**	-	6.993	20.663	11.770	90.000	98.699	90.000	1681.047
**EXPII MH0W**	-	7.099	19.808	11.638	90.000	101.529	90.000	1603.545
**EXPI MH4W**	-	6.965	20.501	11.670	90.000	98.406	90.000	1648.428
**EXPI MH0W**	-	nd	19.790	11.600	nd	nd	nd	nd

**Table 11 molecules-28-07497-t011:** Optimized (DFTB+) unit cell dimensions of the **HH** structures of various hydration ratios, compared with the experimental ones.

Code	Energy [kcal/mol]	a [Å]	b [Å]	c [Å]	α [°]	β [°]	γ [°]	V [Å^3^]
**HH4W**	−256,549.75	30.445	6.159	22.190	89.688	131.558	90.379	3113.379
**HH3W**	−255,972.96	29.973	6.117	22.231	90.169	129.505	89.842	3144.949
**HH2W BLUR**	−253,402.52	30.116	6.037	22.244	90.091	128.996	89.916	3143.336
**HH2W ULBL**	−250,832.71	30.111	5.980	22.222	90.000	128.342	90.000	3138.280
**HH2W ULUR**	−250,833.16	30.167	5.951	22.231	90.182	128.230	89.853	3135.101
**HH1W**	−250,832.65	30.163	5.962	22.235	90.000	128.343	90.000	3136.250
**HH0W**	−256,549.75	30.445	6.159	22.190	89.688	131.558	90.379	3113.379
**EXP HH4W**	-	29.921	6.138	22.041	90.000	128.684	90.000	3160.049

## Data Availability

Data can be obtained from the corresponding author (Ł.S.) by email.

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
