# Peer review of "Density Functional Theory and Density Functional Tight Binding Studies of Thiamine Hydrochloride Hydrates"

_molecules, 2023, doi:10.3390/molecules28227497_

Round 1

Reviewer 1 Report

Comments and Suggestions for Authors

In this article, the authors present a study to characterize, employing quantum chemical calculations, the behavior of and explain the previously obtained NMR spectroscopic results at the molecular level. The study promotes an understanding of the depth of thiamine hydrates and NMR spectra employing dft and dftb. The idea of this work is a novelty; the subject of study is important, and the reported results are reliable, which may provide some guides to further experiments. I thus recommend publication after the following minor revisions.

There are some types:

molecule line 181

amongthose line 328

It is known that temperature affects the optical spectra. However, the authors in line 224 stat the influence of temperature is neglected. Previous woks on molecules mentioned that optical spectra are strongly influenced by temperature, and some explanation should be annexed in a paragraph taking into account. Molecules 2021, 26(18), 5710 Chem. Sci. 2014, 5, 2605–2624 Molecules 2021, 26(13), 3953 Materials 2021, 14, 112.

Comments on the Quality of English Language

There are some types:

molecule line 181

amongthose line 328

check all manuscripts 

Author Response

Comment:

In this article, the authors present a study to characterize, employing quantum chemical calculations, the behavior of and explain the previously obtained NMR spectroscopic results at the molecular level. The study promotes an understanding of the depth of thiamine hydrates and NMR spectra employing dft and dftb. The idea of this work is a novelty; the subject of study is important, and the reported results are reliable, which may provide some guides to further experiments. I thus recommend publication after the following minor revisions.

Response:

Thank you very much for the significant effort needed to create this review. We’ve found all of your suggestions very helpful in improving the quality of our manuscript. Below, please find the direct responses to your comments.

Comment:

There are some types:

molecule line 181

amongthose line 328

Response:

Those typos have been corrected.

Comment:

It is known that temperature affects the optical spectra. However, the authors in line 224 stat the influence of temperature is neglected. Previous woks on molecules mentioned that optical spectra are strongly influenced by temperature, and some explanation should be annexed in a paragraph taking into account. Molecules 2021, 26(18), 5710 Chem. Sci. 2014, 5, 2605–2624 Molecules 2021, 26(13), 3953 Materials 2021, 14, 112.

Response:

The suggested works have been cited in the revised version of the manuscript.

Reviewer 2 Report

Comments and Suggestions for Authors

The manuscript was well written with high-quality results properly described. Only few issues were detected which can be fixed during the proof stage of the manuscript:

1. Line 35: Please replace "anhydrates" with "anhydrides" here and everywhere in the manuscript.

2. Line 92: Please write "e.g." in italics here and everywhere in the manuscript.

3. Line 106: Please write "i.e." in italics here and everywhere in the manuscript.

4. Line 163: "moleculeto" should be replaced with "molecule" here and everywhere in the manuscript.

5. Line 225: "ab initio" should be written here and everywhere in the manuscript in italics. 

Comments on the Quality of English Language

Only few issues were detected. The manuscript was well written. 

Author Response

Comment:

The manuscript was well written with high-quality results properly described. Only few issues were detected which can be fixed during the proof stage of the manuscript:

Response:

Thank you very much for the significant effort needed to create this review. We’ve found all of your suggestions very helpful in improving the quality of our manuscript. Below, please find the direct responses to your comments.

Comment:

  1. Line 35: Please replace "anhydrates" with "anhydrides" here and everywhere in the manuscript.

Response:

We would like to leave the “anhydrates” and do not replace it with “anhydrides” due to the different meaning of those two terms. Anhydrides are compounds that have two acyl groups bonded to the same oxygen atom, while anhydrates are the anhydrous forms of a normally hydrated compounds. This second term is, in our case, the proper one.

Comment:

  1. Line 92: Please write "e.g." in italics here and everywhere in the manuscript.

Response:

This has been corrected, as suggested by the Reviewer.

Comment:

  1. Line 106: Please write "i.e." in italics here and everywhere in the manuscript.

Response:

This has been corrected, as suggested by the Reviewer.

Comment:

  1. Line 163: "moleculeto" should be replaced with "molecule" here and everywhere in the manuscript.

Response:

This has been corrected, as suggested by the Reviewer.

Comment:

  1. Line 225: "ab initio" should be written here and everywhere in the manuscript in italics. 

Response:

This has been corrected, as suggested by the Reviewer.

Reviewer 3 Report

Comments and Suggestions for Authors

In this study, the author conducted computational investigations into the stability of Thiamine hydrochloride under varying levels of hydration. The research involved an extensive review of the literature and the development of a computational model to represent the interaction between water and Thiamine hydrochloride. The obtained results were subsequently validated against experimental data. The introduction is well-crafted, and the computational methodology is adequate.

However, before publication in Molecules, the author should address a few concerns:

  1. In Table 4 and Table 5, it would be more informative to include relative energies rather than absolute energies. This would offer a clearer understanding of the energy changes associated with the removal of water.
  2. In Figure 4, the author might consider consolidating the presentation of parameters a, b, c, and volume into one or two plots instead of individual plots for each parameter.

  1. In Figure 4, there seems to be a discrepancy regarding the Expt II structure's water content, with two green points visible in each plot. Clarification is needed to resolve this issue.

  1. Tables 8 and 9 present NMR data, and it's somewhat challenging to make a direct comparison between them. Simplifying the presentation of this data, perhaps by calculating averages or employing a similar approach, would enhance clarity.

  1. To streamline the main text, the author is encouraged to include only the most relevant graphs. Supplementary Information can be used for figures such as Figure 12 and 13, where individual plots of parameters a, b, and c may not provide substantial additional insights. Focusing on the unit cell volume alone should suffice for the main text
Comments on the Quality of English Language

English is fine. Minor editing might be required.

Author Response

Comment:

In this study, the author conducted computational investigations into the stability of Thiamine hydrochloride under varying levels of hydration. The research involved an extensive review of the literature and the development of a computational model to represent the interaction between water and Thiamine hydrochloride. The obtained results were subsequently validated against experimental data. The introduction is well-crafted, and the computational methodology is adequate.

However, before publication in Molecules, the author should address a few concerns:

Response:

Thank you very much for the significant effort needed to create this review. We’ve found all of your suggestions very helpful in improving the quality of our manuscript. Below, please find the direct responses to your comments.

Comment:

  1. In Table 4 and Table 5, it would be more informative to include relative energies rather than absolute energies. This would offer a clearer understanding of the energy changes associated with the removal of water.

Response:

Columns presenting the relative energies have been added, as suggested by the Reviewer.

Comment:

  1. In Figure 4, the author might consider consolidating the presentation of parameters a, b, c, and volume into one or two plots instead of individual plots for each parameter.

Response:

We initially thought that it would be better, but due to the significant differences in the values of the unit cell parameters, the observed changes are better visible when the data are presented the way they are now.

Comment:

  1. In Figure 4, there seems to be a discrepancy regarding the Expt II structure's water content, with two green points visible in each plot. Clarification is needed to resolve this issue.

Response:

In the work 10.1021/cg0340749, abbreviated in this article as EXPII (which was explained in Table 1), the authors have recorded two structures of MH, with either 4 or 0 molecules of water in the unit cell. Those information are available also in Table 4. Therefore, those two green points represent those two structures. We have used the same color to indicate that they originate from the same work (10.1021/cg0340749).

Comment:

  1. Tables 8 and 9 present NMR data, and it's somewhat challenging to make a direct comparison between them. Simplifying the presentation of this data, perhaps by calculating averages or employing a similar approach, would enhance clarity.

Response:

The data presented in Tables 8 and 9 do not need to be compared as those tables present the results obtained for different hydrates. Calculating the averages would extend those already large tables, and, in our opinion, wouldn’t be very informative.

Comment:

  1. To streamline the main text, the author is encouraged to include only the most relevant graphs. Supplementary Information can be used for figures such as Figure 12 and 13, where individual plots of parameters a, b, and c may not provide substantial additional insights. Focusing on the unit cell volume alone should suffice for the main text

Response:

Figures 12 and 13 have been moved to the Supplementary Information file, as suggested by the Reviewer.